# Optimization Design and Test of Spike-Toothed Crop Divider

**Zhengliang Ding** [1]**, Shaochun Ma** [1,2,]*****, Jing Bai** [1]**, Wenpeng Liang** [3] **and Xiadong Zhang** [3]

[1] Beijing Key Laboratory of Optimized Design for Modern Agricultural Equipment, College of Engineering, China Agricultural University, Beijing 100083, China

[2] Guangxi Academy of Sciences, Nanning 530007, China

[3] Guangxi Agricultural Machinery Research Institute Co., Ltd., Nanning 530007, China

***** Correspondence: shaochun2004@cau.edu.cn; Tel.: +86-010-6273-6638

**Abstract:** The crop divider is a crucial component of the sugarcane chopper harvester that has an important effect on the lifting performance of lodged sugarcanes. However, tropical weather and climate result in severe sugarcane lodging, which seriously hinders the sugarcane lifting process. To improve the lifting performance of crop dividers, a variable-spiral spike-toothed crop divider was designed. Some structural parameters of the bench test were designed. In this study, the forward speed, the rotational speed of the scrolls, and the installation angle of the scrolls were selected as test factors and the lifting angle of the sugarcane was selected as the test index. Based on a single-factor test, we found the optimum range of these test factors: a forward speed of 0.4~0.8 m/s, a rotational speed of the inside scrolls of 100~140 r/min, and an installation angle of the inside scrolls of 55°~65°. An orthogonal test was conducted to identify the relationships of test index and test factors. The regression model was obtained by fitting the test data and the response surface was established to analysis the interaction of test factors. According to Design Expert 8.0.6, the optimal factor combination of parameters was: a forward speed was 0.74 m/s, a rotational speed of the inside scrolls of 134 r/min, and an installation angle of the inside scrolls of 63.5°. Finally, the regression model was verified by the bench test. The relative error of the regression model of the lifting angle was 4.94%, which showed that the regression model was reliable. This study is expected to provide valuable references for the design and improvement of crop dividers.

**Keywords:** sugarcane; lifting performance; crop divider; spike tooth; lifting angle

## 1. Introduction

Sugarcane, which is widely planted in the south of China, is an important raw material for sugar production. The mechanized harvesting of sugarcane is seriously restricted by the planting mode and terrain [1–3]. Due to the high cost of manual harvesting, it is necessary to develop sugarcane mechanization [4–8]. The crop divider is a crucial component of sugarcane harvesting; the intertwined stalks are separated and lifted by the crop divider [9,10]. During harvesting, sugarcane is often pushed down or slips from the crop divider, resulting in serious waste. The lifting quality of a crop divider is directly related to whether the sugarcane can be fed into the basecutter smoothly [11–13]. Therefore, it is of great significance to improve the lifting quality of crop dividers.

There have been some studies on the structure of crop dividers. Deng et al. [14] proposed that adding a conical spiral at the front end of crop dividers had a greater lifting effect on the root of sugarcane, which was conducive to the lifting of severely lodged sugarcane. Dong et al. [15] proposed a design method for a variable-spiral crop divider, established a physical model, and verified the rationality of the structure through a simulation. Gao et al. [16] established a physical model for a sugarcane harvester by using finite element technology and discussed the action process between the crop divider and sugarcane. Zhang et al. [17] designed a finger-chain-type crop divider; the regularities of structural parameters and motion parameters on sugar-lifting were studied through virtual

orthogonal tests and virtual two-factor tests. Song et al. [18] designed a two-stage spiral lifting and picking-up mechanism and verified the lifting effect through a simulation and indoor experiments. Xie et al. [19] designed a lifting device that combined spiral scrolls and a chain lifter; the researchers used experiments and a high-speed-camera analysis to verify the lifting quality. The lifting quality of sugarcane is affected by the leaf shape on the surface of the crop divider [20–22]. At present, the sugarcane crop divider is mostly formed by a spiral catheter. In the process of sugarcane lifting, the point contact between the sugarcane and spiral blade makes it easy for sugarcane to slip.

Through the study of existing crop dividers and an analysis of lodged sugarcane, we designed a variable-spiral spike-toothed crop divider. In this study, the forward speed, the rotational speed of the scrolls, and the installation angle of the scrolls were selected as test factors and the lifting angle of the sugarcane was selected as the test index. The indoor test was conducted to analysis the interaction of test factors.

## 2. Materials and Methods

### 2.1. Structure and Principle of Crop Dividers

To improve the lifting performance of the crop divider and reduce sugarcane slip, a spike-toothed crop divider of was designed in this study. As shown in Figure 1, the crop divider test platform consisted of a hydraulic motor, an upper bracket, spike teeth, a spiral scroll, a lower bracket, a left bracket, a right bracket, and a bottom base. The spiral scroll was installed on the left bracket through the upper and lower bracket. A row of positioning pin holes was punched on the side of the right bracket, which was composed of two sections. The hydraulic motor was installed on the upper end of the spiral scroll. A bearing was installed between the scroll and the bracket; the length of the right bracket could be adjusted through the pin hole. The hydraulic motor was installed on the spiral scroll through the bearing and the spikes were welded to the surface of the spiral scroll in a spiral manner. The main technical parameters of the test bed are shown in Table 1.

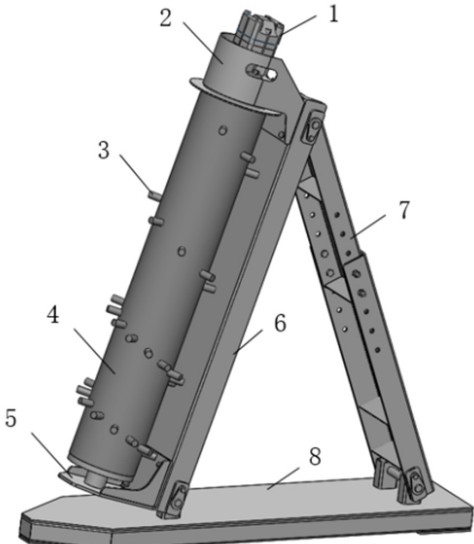

**Figure 1.** Structure of test bed for crop divider. 1. Hydraulic motor. 2. Upper bracket. 3. Spike tooth. 4. Spiral scroll. 5. Lower bracket. 6. Left bracket. 7. Right bracket. 8. Bottom base.

**Table 1.** Main technical parameters.

| Parameters | Value |
| --- | --- |
| Entire machine size (mm × mm × mm) | 1500 × 300 × 1800 |
| Matched power (kW) | ≥103 |
| Forward speed (m/s) | 0~2 |

The power of the variable-spiral spike-toothed crop divider was provided by the hydraulic station, which drove the hydraulic motor to work, and the hydraulic motor, which drove the spiral scroll to rotate. When the lodged sugarcane stem contacted the crop divider, it was supported to move upward by the spiral spikes. The spikes held the sugarcane to prevent it from slipping. As the scroll rotated spirally, the spikes drove the sugarcane upward. When the lodged sugarcane was gradually lifted up, the distance between the sugarcane root and the roller gradually decreased, which made the angle between the sugarcane stem and the ground gradually increase. Therefore, the spiked teeth were arranged on the surface of the scroll in a variable-spiral manner, which could effectively improve the lifting efficiency of the sugarcane and increase the lifting angle of the sugarcane. The thrust of the spiked teeth on the spiral scroll had an important influence on the lifting performance of the sugarcane. The lifting process of the variable-spiral spike-toothed crop divider is presented in Figure 2. As shown in Figure 2a, when the sugarcane and the scroll were just in contact, the sugarcane was in contact with multiple spikes at the same time with a large contact area. Figure 2b shows the rising process of the sugarcane and the number of spikes in contact with the sugarcane gradually decreasing; Figure 2c shows the late stage of the sugarcane lifting when the sugarcane only touched one tooth. Therefore, in the process of lifting the sugarcane, the spiral thrust of a cluster of spikes was gradually transformed into the upward-pushing force of the spikes on the sugarcane.

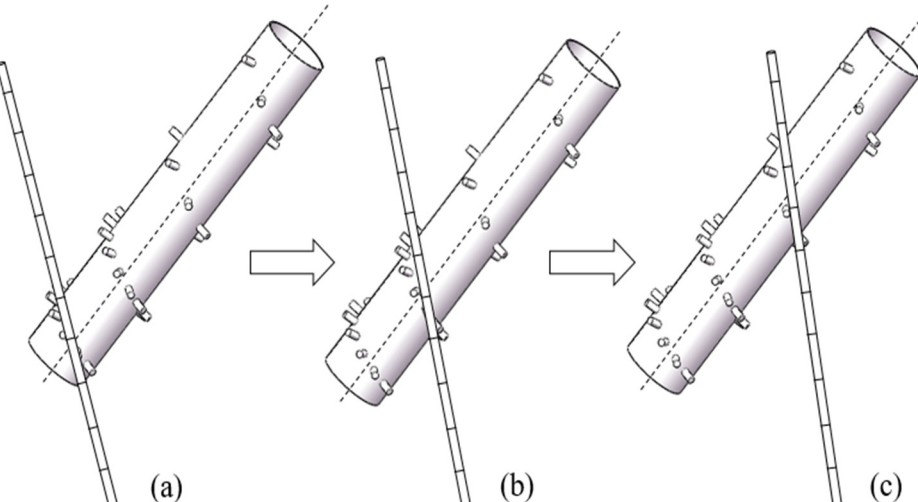

**Figure 2.** Lifting process of spike-toothed crop divider: (**a**) the sugarcane and the scroll were just in contact, (**b**) the rising process of the sugarcane, (**c**) the late stage of the sugarcane lifting.

### 2.2. Design of Structural Parameters
Design of Spiral Scroll

If the diameter of the spiral scroll was too small, the scroll would be easily wound by the sugarcane leaves; if the diameter was too large, the scroll would be bulky and the processing cost would be high. Therefore, the diameter of the scroll had to meet the following conditions:

$$D \geq \frac{\sqrt{\left(k^2 - S^2\right)}}{\pi} \tag{1}$$

where $k$ = the length of sugarcane leaf (600 mm) and $S$ = pitch (210 mm).

The scroll diameter was calculated to be not less than 178.91 mm; therefore, we selected 200 mm.

The length of the spiral scroll was closely related to the lifting height of the sugarcane. To prevent the sugarcane from slipping, it was necessary to ensure that the sugarcane could

be lifted to a certain height that had to be greater than the height of the center of gravity of the sugarcane. Therefore, the scroll length had to meet the following inequality:

$$L \geq \frac{l\sin\delta}{2\sin\gamma} \tag{2}$$

where $l$ = sugarcane stalk length (3000 mm); $\gamma$ = the installation angle of the crop divider (60°); and $\delta$ = the maximum angle between the sugarcane stem and the ground (65°).

We calculated that the length of the scroll could not be less than 1570 mm. Because a distance had to be reserved between the upper and lower ends of the spiral scroll, we selected 1600 mm as the length of the scroll.

### 2.3. Design of Spiral Line

The variable-spiral spike-toothed crop divider was designed to prevent the sugarcane from slipping due to the increasing lift angle. As shown in Figure 3, the spiral line was formed by the rotation of a point on the bus line with the Z-axis and the moving point was accelerated upward at the same time. The trajectory of the moving point ($k1$, $k2$....., $kn$) was a parabola. Figure 3a presents a schematic diagram of the formation of the variable spiral. Figure 3b shows an expanded view of a variable spiral. Figure 3c shows the reference line of the spiked teeth on the surface of the scroll from the top view.

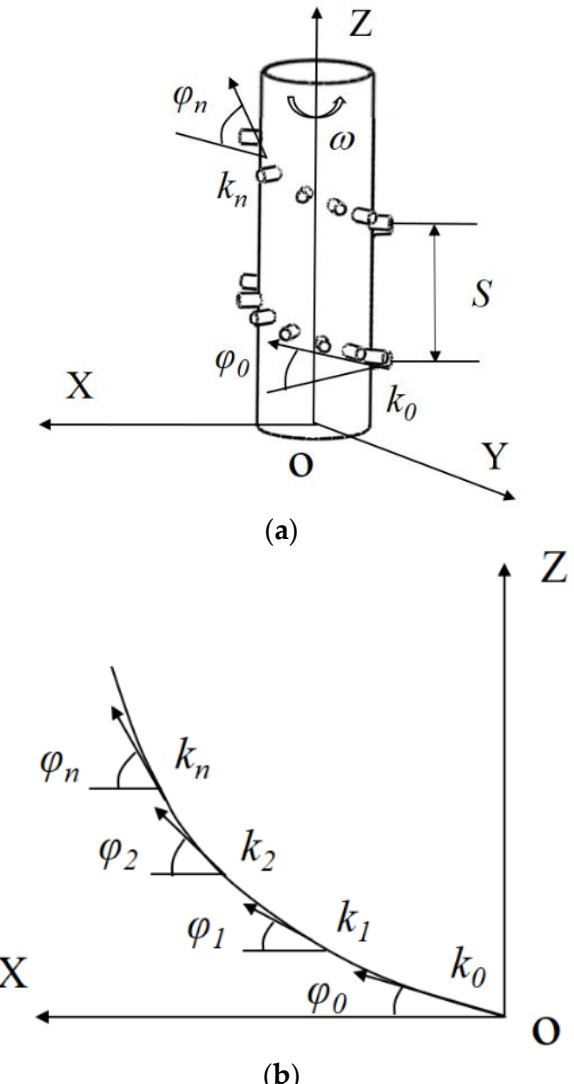

**(a)**

**(b)**

**Figure 3.** *Cont.*

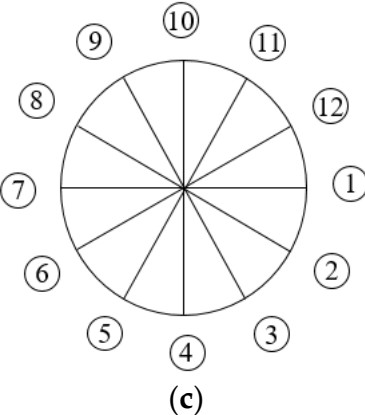

(c)

**Figure 3.** Spiral analysis: $k_n$ = a moving point on a spiral; $\varphi_n$ = spiral angle; $S$ = spiral pitch; $\omega$ = angular velocity of scroll; (**a**) Formation of a variable spiral, (**b**) Expanded view of a variable spiral, (**c**) Reference line of spiked teeth.

The mathematical equation of the variable spiral is given as:

$$\begin{cases} x = R\cos\omega t \\ y = R\sin\omega t \end{cases} \tag{3}$$

where $R$ = radius of the scroll (mm); $\omega$ = angular velocity of the scroll (rad/s); and $t$ = rotation time of the scroll (s).

The parabolic equation of the expanded view of a variable spiral is given as:

$$z = ax^2 + bx \tag{4}$$

where $a$ = the quadratic coefficient of the parabola and $b$ = the first-order coefficient of the parabola.

The unfolded parabola was derived to obtain the slope at any point on the parabola, which was the tangent of the spiral angle:

$$\tan\varphi_n = 2ax + b \tag{5}$$

where $\varphi_n$ = spiral rise angle at any point on the spiral (in degrees).

When the spiral was in the initial position, $b = \tan\varphi_0$, the selection of the spiral rise angle had to meet the requirements of the sugarcane lodged angle. Therefore, the initial spiral rise angle was taken as 20°. The quadratic coefficient of the parabolic equation was:

$$a = \frac{h_n - x\tan\varphi_0}{x^2} \tag{6}$$

where $h_n$ = the height of any point on the spiral (mm).

The number of spiral lines was 4. At the top of the spiral lines, the abscissa was 4 times the circumference of the scrolls. The effective height of the spiked spiral lines was 1400 mm. Therefore, the parabolic equation was obtained as:

$$z = 7.8\times10^{-5}x^2 + 0.36x \tag{7}$$

The spiral rise angle at any point on the spiral lines was obtained as:

$$\varphi_n = \text{aratan}\left(\frac{2h_n}{x} - \tan\varphi_0\right) \tag{8}$$

According to the actual harvesting situation of the sugarcane crop divider, the above data were brought into the formula to calculate the spiral rise angle of the upper end of the scroll, which was 40°.

### 2.4. Design of Spiked Teeth

The material used for the spiked teeth was steel rods; these were arranged in a variable-spiral manner and welded on the surface of the spiral scroll. To ensure the support strength of the sugarcane, we selected 15 mm as the diameter of the spiked teeth.

The spiked teeth had a supporting effect on the sugarcane: the length of the spiked teeth had an important relationship with the lifting of the sugarcane. In the early stage of lifting, the spiked teeth were too long to be inserted into the root of the sugarcane; if the spiked teeth were too short, the sugarcane easily slipped off. When the sugarcane was lifted up by the spikes, it was supported by three spikes at the same time, so the lifting process was relatively stable, as shown in Figure 4. Therefore, the calculation formula of the tooth length $j$ is given as:

$$j \geq \frac{R+d}{\cos30} - R \tag{9}$$

where $d$ = diameter of the sugarcane (30 mm).

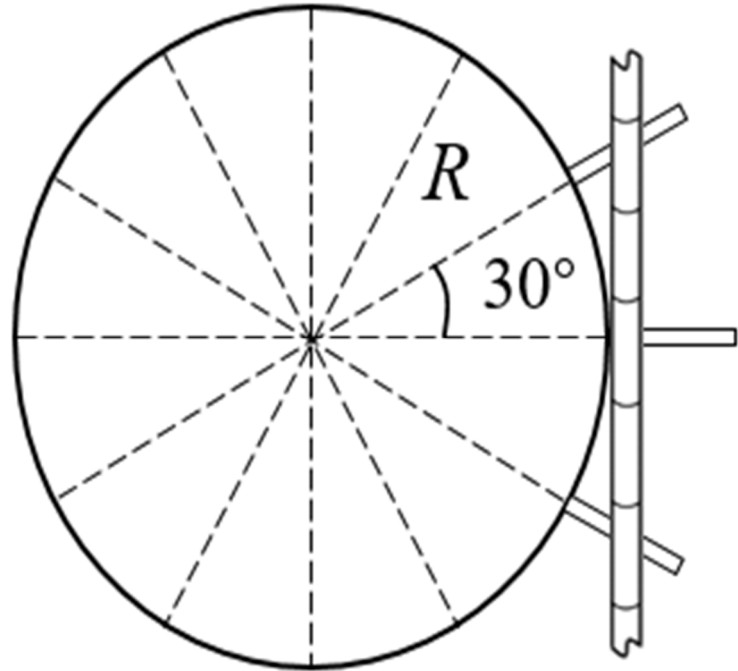

**Figure 4.** Contact between spiked teeth and sugarcane (top view of scrolls).

The length of the spiked teeth was calculated to be no less than 50 mm; a part of the safety length was reserved, which we selected to be 80 mm.

### 2.5. Design of Bracket

A triangle was formed between the left brackets, right brackets, and the base, as shown in Figure 5. The length of the left brackets and the base was fixed and the length of the right brackets had to meet the installation angle range of the spiral scroll. Therefore, the calculation formula of the right brackets ($e$) was as follows:

$$e = \sqrt{(L\sin\gamma)^2 + (u-(L\cos\gamma))^2} \tag{10}$$

where $u$ = the length of base (1000 mm) and $\gamma$ = the installation angle of the scroll (55°~75°).

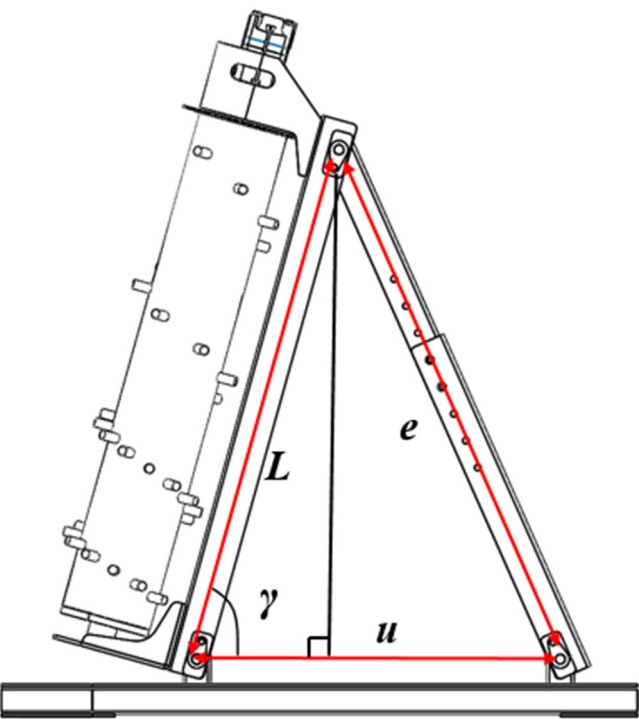

**Figure 5.** Shelf bracket.

According to the formula, the length of the right brackets was 1312~1652 mm. According to the calculated length, a locating pin hole was punched on the side of the right brackets to adjust the length of these brackets, which changed the installation angle of the scroll.

### 2.6. Experiment Material

The test was carried out in January 2021 at Guangxi Agricultural Machinery Research Institute Co., Ltd, Guangxi Zhuang Autonomous Region, China. The sugarcane was obtained from Wuming County, Nanning City, Guangxi Zhuang Autonomous Region, China. The fresh sugarcane that was cut from the field was perennial sugarcane in the first year. The planting parameters are shown in Table 2. Sugarcane stems with relatively straight growth were selected to avoid the influence of other factors. Before the test, the sugarcane was stripped and pieces with roughly the same effective length and diameter were selected.

**Table 2.** Sugarcane planting parameters.

| Plant Spacing/m | Moisture Content/% | Average Diameter/mm | Average Height/mm | Number of Internodes/Number |
|:---:|:---:|:---:|:---:|:---:|
| 1.2 | 29.71 | 27.9 | 3420 | 18 |

According to the growing conditions of the sugarcane in the field, the sugarcane was clamped with a clamp to adjust it to the corresponding lodging posture.

### 2.7. Experiment Method

The lifting angle refers to the angle between the sugarcane stalk and the ground when the sugarcane was lifted to the highest point. The lifting angle could be used to measure

the lifting effect of the crop divider, which was calculated using the following formula (Song et al., 2012):

$$\tan\delta = \frac{h}{\sqrt{l_1^2 + l_2^2}} \tag{11}$$

where $h$ = the lifting height of sugarcane (mm); $l_1$ = the vertical distance between the sugar cane row and crop divider (mm); and $l_2$ = the distance between the contact point and the root of the sugarcane in the direction of the sugarcane row (mm).

As shown in Figure 6, a cartesian coordinate system was established. A sticky ruler was pasted on the sugarcane crop divider and a high-speed camera was used to read the scale when the sugarcane was held to the highest point. The value of the lifting angle of the sugarcane was obtained by measuring the values of $h$ and $l_2$. During the test, the serious lodging of sugarcane was simulated: the lodging angle $\alpha$ of the sugarcane was $10°\sim20°$ and the slip angle $\beta$ was $80°\sim100°$ (Lin et al., 2012). The motor drove the trolley to move forward and the time was recorded using a stopwatch to calculate the forward speed. The position of the pin hole of the scroll bracket was adjusted to obtain different bracket lengths, which could adjust different scroll angles. The flow of the hydraulic motor was adjusted to obtain different scroll rotational speeds by installing the hydraulic speed control valve.

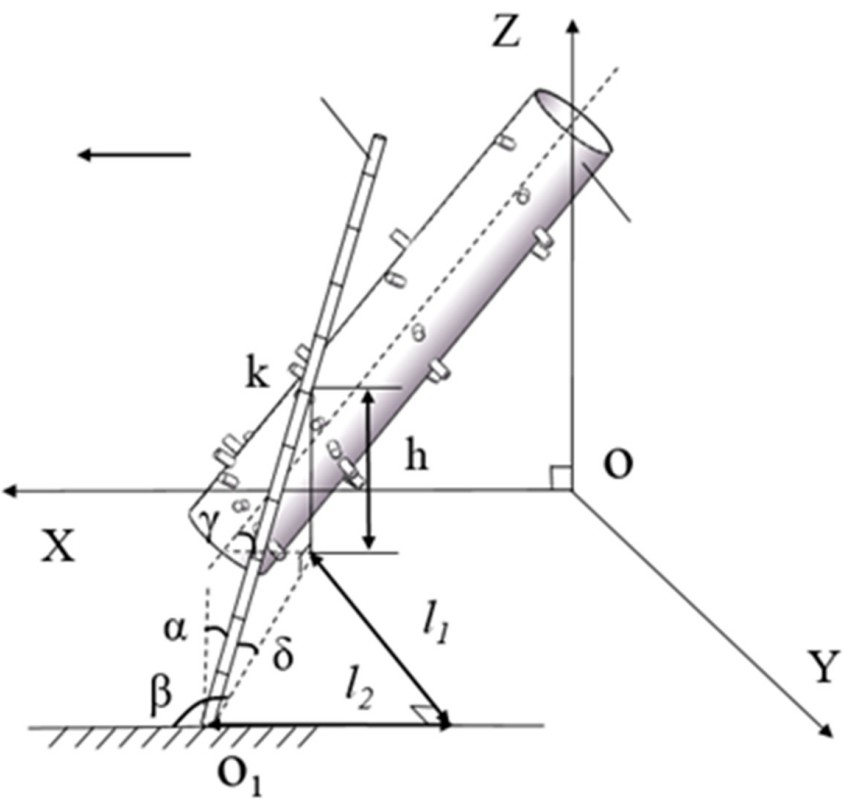

**Figure 6.** Schematic diagram of sugarcane lifting: $\alpha$ = lodging angle; $\beta$ = side angle; $\gamma$ = installation angle of scroll; $\delta$ = lifting angle; $k$ = contact point between sugarcane and scroll.

In this study, the forward speed, the rotational speed of the scrolls, and the installation angle of the scrolls were selected as test factors, the lifting angle of sugarcane was selected as test index. The single factor test and the orthogonal test was conducted to identify the relationships of test index and test factors. The test was repeated 3 times to reduce the experimental error for each lodging angle, and the results were averaged.

## 3. Results

### 3.1. Sugarcane Lifting Performance Test

Single-Factor Test

In the single-factor test, the forward speed, the rotational speed of the scrolls, and the installation angle of the scrolls were selected as the test factors, and the lifting angle of the sugarcane was selected as the test index. According to the actual production and operation experience, for the test range of the single-factor test: the forward speed was 0.2–1.0 m/s, the rotational speed of the inside scrolls was 80–160 r/min, and the installation angle of the inside scrolls was 55–75°. During the experiment, two test factor levels were fixed in each group of experiments. The change law of the sugarcane lifting angle was explored with the linear change in the single-factor test. To reduce the test errors, each group of tests was repeated three times and the results were taken as the average. The test results are shown in Figure 7.

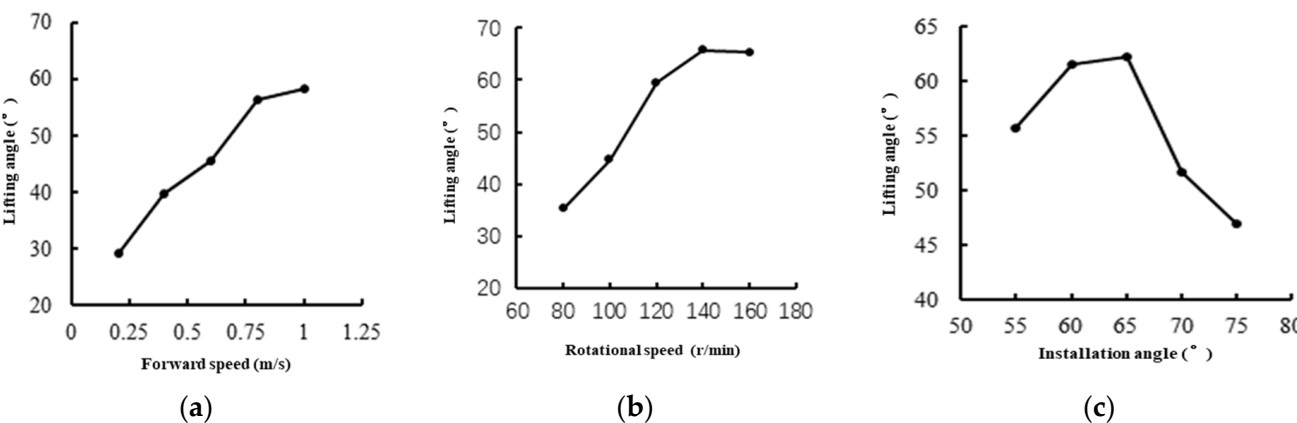

**Figure 7.** Results of single-factor test: (**a**) Effect of forward speed on lifting angle, (**b**) Effect of rotational speed on lifting angle, (**c**) Effect of installation angle on lifting angle.

During the test, different forward speeds were used with a rotational speed of 120 r/min and an installation angle of 60°. As shown in Figure 7a, the lifting angle of lodged sugarcane had an overall upward trend with an increase in the forward speed. The lifting angle had an obvious upward trend in the range of 0.4~0.8 m/s. When the forward speed was low, the lifting efficiency also was low. In the later stage of lifting, as the spiral rise angle of the arranged spiked teeth gradually increased, the lifting height increased in per-unit time, which caused the lifting angle of the sugarcane to also increase.

During the test, different rotational speeds were used with a forward speed of 0.6 m/s and an installation angle of 60°. As shown in Figure 7b, the lifting angle of the lodged sugarcane had an overall upward trend with an increase in the rotational speed. The lifting angle had an obvious upward trend in the range of 100~140 r/min. When the rotational speed of the scroll increased, the sugarcane lifting angle also increased.

During the test, different installation angles were used with a forward speed of 0.6 m/s and a rotational speed of 120 r/min. As shown in Figure 7c, with an increase in the the installation angle of the scrolls, the lifting angle of the lodged sugarcane showed a trend of first rising and then falling. Therefore, in the range of 55°~65°, the lifting angle of the sugarcane could be effectively improved by increasing the installation angle of the scrolls.

### 3.2. Orthogonal Test

An orthogonal test was carried out to further study the lifting performance of the lodged sugarcane. Based on the single-factor test, we found the following optimum ranges of these test factors: the forward speed was 0.4~0.8 m/s, the rotational speed of the inside scrolls was 100~140 r/min, and the installation angle of the inside scrolls was 55°~65°. The orthogonal test was conducted to identify the relationships of the test index and the test

factors. Each group of experiments was repeated three times and the results were averaged. The level coding of each test factor is shown in Table 3 and the orthogonal test scheme and results are shown in Table 4.

**Table 3.** Factors and levels of orthogonal tests.

| Levels | Test Factors | | |
|---|---|---|---|
| | Forward Speed $x_1$/m/s | Rotational Speed $x_2$/r/min | Installation Angle $x_3$/° |
| $-\gamma$ | 0.40 | 100 | 55.0 |
| $-1$ | 0.46 | 106 | 56.5 |
| 0 | 0.60 | 120 | 60.0 |
| 1 | 0.74 | 134 | 63.5 |
| $\gamma$ | 0.80 | 140 | 65.0 |

Note: $\gamma$ represents the distance between the asterisk point and central point ($\gamma$ = 1.414).

**Table 4.** Orthogonal test scheme and results.

| Numbers | Test Factors | | | Lifting Angle $y$ /° |
|---|---|---|---|---|
| 1 | $-1$ | $-1$ | $-1$ | 48.35 |
| 2 | 1 | $-1$ | $-1$ | 59.71 |
| 3 | $-1$ | 1 | $-1$ | 54.42 |
| 4 | 1 | 1 | $-1$ | 66.80 |
| 5 | $-1$ | $-1$ | 1 | 59.96 |
| 6 | 1 | $-1$ | 1 | 61.03 |
| 7 | $-1$ | 1 | 1 | 68.84 |
| 8 | 1 | 1 | 1 | 70.00 |
| 9 | $-\gamma$ | 0 | 0 | 55.83 |
| 10 | $\gamma$ | 0 | 0 | 68.75 |
| 11 | 0 | $-\gamma$ | 0 | 42.16 |
| 12 | 0 | $\gamma$ | 0 | 55.31 |
| 13 | 0 | 0 | $-\gamma$ | 43.73 |
| 14 | 0 | 0 | $\gamma$ | 62.97 |
| 15 | 0 | 0 | 0 | 49.50 |
| 16 | 0 | 0 | 0 | 53.36 |
| 17 | 0 | 0 | 0 | 52.52 |
| 18 | 0 | 0 | 0 | 48.57 |

Based on the orthogonal test results, Design Expert 8.0.6 software was used to perform regression fitting on the test data. The regression model between the lifting height and forward speed, rotational speed, and installation angle of the spiral scroll was obtained as:

$$y = 50 + 3.69x_1 + 4.13x_2 + 4.81x_3 + 0.14x_1x_2 - 2.69x_1x_3 + 0.59x_2x_3 + 7.13x_1^2 + 0.35x_2^2 + 2.66x_3^2 \tag{12}$$

The results of the variance analysis of the regression model are shown in Table 5. The value for this model was less than 0.01, which indicated that it had a strong significance. The interaction term of the forward speed and rotational speed ($x_1x_2$), the interaction between the rotational speed and installation angle ($x_2x_3$), and the quadratic term of the rotational speed ($x_2^2$) had no significant effect on the lifting angle ($p > 0.1$), which was eliminated from the regression model. Other items had a significant impact on the lifting angle ($p < 0.05$) in Table 5. The lack of fit was not significant ($p = 0.2468$), which indicated that there were no other major factors that affected the lifting angle. A new regression model was obtained by optimizing the model and eliminating the insignificant items:

$$y = 50.24 + 3.69x_1 + 4.13x_2 + 4.81x_3 - 2.69x_1x_3 + 7.13x_1^2 + 2.66x_3^2 \tag{13}$$

**Table 5.** Regression model variance analysis.

| Source | Sum of Squares | Freedom | F-Value | p-Value |
|---|---|---|---|---|
| Model | 1171.73 | 9 | 12.82 | 0.0007 |
| $x_1$ | 163.11 | 1 | 16.06 | 0.0039 |
| $x_2$ | 205.07 | 1 | 20.02 | 0.0020 |
| $x_3$ | 278.00 | 1 | 27.38 | 0.0008 |
| $x_1 x_2$ | 0.15 | 1 | 0.015 | 0.9050 |
| $x_1 x_3$ | 57.84 | 1 | 5.70 | 0.0441 |
| $x_2 x_3$ | 2.75 | 1 | 0.27 | 0.6169 |
| $x_1^2$ | 407.03 | 1 | 40.09 | 0.0002 |
| $x_2^2$ | 1.01 | 1 | 0.099 | 0.7609 |
| $x_3^2$ | 56.71 | 1 | 5.59 | 0.0457 |
| Residual | 81.23 | 8 | | |
| Lack of Fit | 65.19 | 5 | 2.44 | 0.2468 |
| Pure Error | 16.03 | 3 | | |
| Cor Total | 1252.96 | 17 | | |

Note: $p < 0.01$ (extreme significance), $p < 0.05$ (significance), $p > 0.05$ (no significance).

### 3.3. Analysis of Response Surface

Based on the optimal parameter combination and the regression model, a two-factor interactive response-surface analysis was carried out to establish the orthogonal test response surface, as shown in Figure 8. Figure 8a shows the response-surface graph of the forward speed and the rotational speed to the lifting angle of the lodged sugarcane when the installation angle of the scroll was 60°. It can be seen in the figure that when the forward speed was 0.60~0.74 m/s and the rotational speed was 120~134 r/min, the lifting angle was larger. When the rotational speed of the scroll was constant, the lifting angle first decreased and then increased with the increase in the forward speed; when the forward speed was constant, the lifting angle increased with the increase in the rotational speed of the scroll. At the bottom of the scroll, the spiral rise angle of the spiked teeth was small; when the forward speed was low, the lodged sugarcane could not be lifted to the upper part of the scroll in time. When the rotational speed of the scroll was fast, the sugarcane rose faster with the scroll and the lifting angle was large.

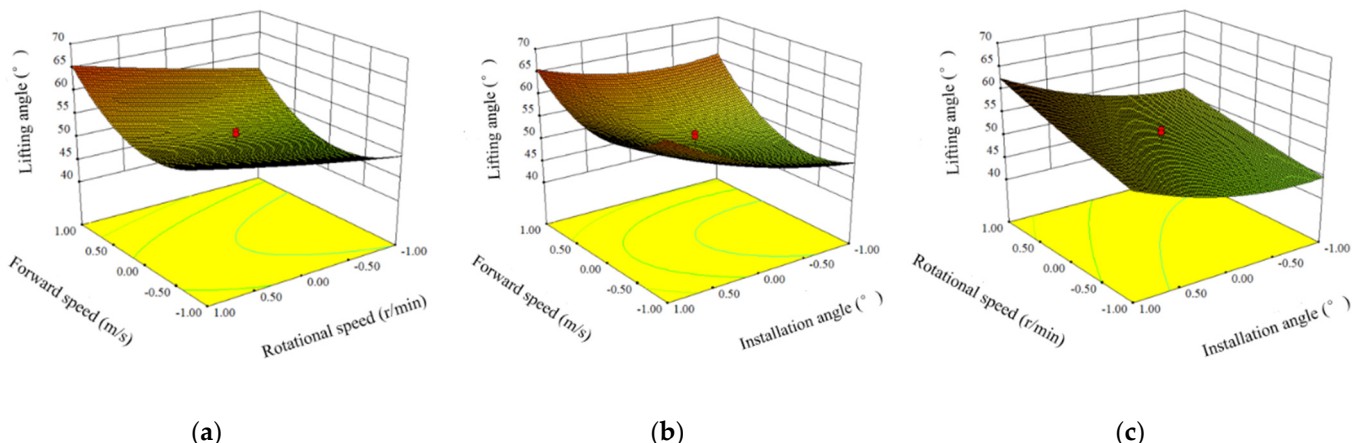

(a)　　　　　　　　　　　　　(b)　　　　　　　　　　　　　(c)

**Figure 8.** Response surface analysis. (**a**) $y = f(x_1, x_2, 0)$; (**b**) $y = f(x_1, 0, x_3)$; (**c**) $y = f(0, x_2, x_3)$.

Figure 8b shows the response-surface graph of the forward speed and the installation angle of the scroll to the lifting angle of the lodged sugarcane when the rotational speed was 120 r/min. It can be seen in the figure that when the forward speed was 0.60~0.74 m/s and the installation angle of the scroll was 56.5°~63.5°, the lifting angle was larger. When the forward speed was constant, the lifting angle increased with the increase in the installation

angle of the scroll; when the installation angle of the scroll was constant, the lifting angle first decreased and then increased with the increase in the forward speed. The larger the installation angle of the scroll, the larger the sugarcane lifting angle with the increase in the overall height of the spiral scroll.

Figure 8c shows the response-surface graph of the rotational speed and the installation angle of the scroll to the lifting angle of the lodged sugarcane when the forward speed was 0.6 m/s. It can be seen in the figure that when the rotational speed was 120~134 r/min and the installation angle was 60°~63.5°, the lifting angle was larger. When the rotational speed of the scroll was constant, the lifting angle increased with the increase in the installation angle of the scroll; When the installation angle of the scroll was constant, the lifting angle increased with the increase in the rotational speed. The increase in the installation angle of the scroll was helpful in lifting the lodged sugarcane, which could be lifted up to the highest point in a short time at a higher rotational speed.

### 3.4. Model Optimization and Verification

According to the optimization module of the Design Expert 8.0.6 software, the maximum response value was set to calculate the corresponding optimization results. The optimal factor combination of parameters was: a forward speed of 0.74 m/s, a rotational speed of the inside scrolls of 134 r/min, and an installation angle of the inside scrolls of 63.5°.

Based on the bench test, a verification test was carried out to further verify the accuracy of the model, as shown in Figure 9. Each group of tests was repeated three times to obtain the average value. The results are shown in Table 6. The test results were consistent with the predicted values and the relative error was small, so the regression model was reliable.

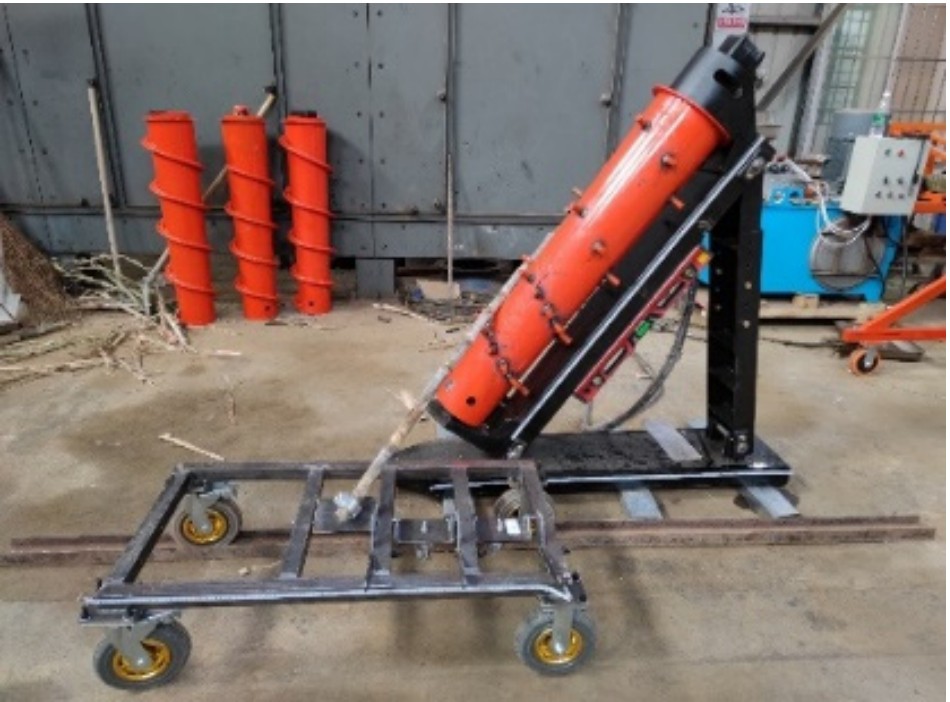

**Figure 9.** Test bed for spike-toothed crop divider.

**Table 6.** Verification results of regression model.

|  | Predictive Value/° | Test Value/° | Relative Errors/% |
| --- | --- | --- | --- |
| Lifting angle | 70.8 | 67.3 | 4.94 |

## 4. Conclusions

In this study, the forward speed, the rotational speed of the scrolls, and the installation angle of the scrolls were selected as the test factors and the lifting angle of the sugarcane was selected as the test index. Based on the single-factor test, the optimum range of these test factors was: a forward speed of 0.4~0.8 m/s, a rotational speed of the inside scrolls of 100~140 r/min, and an installation angle of the inside scrolls of 55°~65°.

An orthogonal test was conducted to identify the relationships of the test index and the test factors. According to Design Expert 8.0.6, the optimal factor combination of parameters was: a forward speed of 0.74 m/s, a rotational speed of the inside scrolls of 134 r/min, and an installation angle of the inside scrolls of 63.5°.

The regression model was verified by the bench test. The relative error of regression model of the lifting angle was 4.94%, which showed that the regression model was reliable.

**Author Contributions:** Conceptualization, Z.D. and S.M.; methodology, Z.D.; software, Z.D. and J.B.; validation, Z.D., X.Z. and W.L.; resources, Z.D.; data curation, Z.D. and J.B.; writing—original draft preparation, Z.D.; writing—review and editing, Z.D.; project administration, S.M.; funding acquisition, S.M. All authors have read and agreed to the published version of the manuscript.

**Funding:** This research was funded by Portable Sugarcane Harvester Research and Development (NK2022160504), the National Natural Science Foundation of China (Grant No. 32071916), Study on mechanism of efficiently removing sugarcane impurities (322CXTD521), the 2115 Talent Development Program of China Agricultural University, Yunnan Zhenkang Professor Workstation Grant, Zhenkang sugarcane mechanized harvesting project. Any opinions, findings, and conclusions expressed in this publication are those of the authors and do not necessarily reflect the view of China Agricultural University.

**Institutional Review Board Statement:** Not applicable.

**Informed Consent Statement:** Not applicable.

**Data Availability Statement:** All relevant data presented in the article are stored according to institutional requirements and, as such, are not available online. However, all data used in this manuscript can be made available upon request to the authors.

**Conflicts of Interest:** The authors declare no conflict of interest.

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
