# Peer review of "Optimization Design and Test of Spike-Toothed Crop Divider"

_agriculture, doi:10.3390/agriculture12101692_

Round 1

Reviewer 1 Report

Overall, this paper is in a good shape and the study can provide a valuable reference in crop divider development. The test and analysis procedures are detailed and reliable. The specific comments is as follows:

1. Page 3 line 105: The crop divider has variable spiral spike –tooth. Why was the Pitch 210 mm ?

2. In Section 2.4, what is the left bracket length L value ?

3. Page 6 line 183: The specific arrangement of the positioning pin holes on the side of the right bracket needs to be introduced. The position of the positioning pin holes will directly limit the adjustment range and specific value of the installation inclination.

4. Page 7 line 196: According to the growing conditions of the sugarcane in the field, the sugarcane was clamped with a clamp to adjust to the corresponding lodging posture. Please describe the "corresponding lodging posture" in detail.

Author Response

Dear Editor,

Based on the constructive comments, edits and suggestions provided by you and the reviewers, we have conducted a thorough revision of the manuscript. We have responded to all edits, comments and suggestions as discussed in the following pages in this document.

Please let me know if you have any questions or need further clarification.

Thank you very much for your time processing the manuscript.

Sincerely,

Shaochun Ma, Ph.D

Associate Professor

College of Engineering

China Agricultural University

No.17 Qinghua East Road, Haidian District, Beijing, 100083

Ph: 010-62736638; Cell: 18210198629

Email:[email protected]

Reviewer #1: Overall, this paper is in a good shape and the study can provide a valuable reference in crop divider development. The test and analysis procedures are detailed and reliable. The specific comments is as follows:

1: Page 3 line 105: The crop divider has variable spiral spike –tooth. Why was the Pitch 210 mm ?

Authors: Thanks for the valuable comments.. When designing the diameter of the spiral scroll, the circumference within a pitch of the spiral scroll shall not be less than the length of sugarcane leaves to avoid winding. Therefore, the pitch is taken as a minimum of 210mm.

2: In Section 2.4, what is the left bracket length L value ?

Authors: We are so grateful for your kind question. The L value of the left bracket length is 1600mm.

3: Page 6 line 183: The specific arrangement of the positioning pin holes on the side of the right bracket needs to be introduced. The position of the positioning pin holes will directly limit the adjustment range and specific value of the installation inclination.

Authors: Thanks for the valuable comments. The right bracket is divided into two parts, five holes are drilled respectively, and the length adjustment range of the right bracket is 1312-1652mm. When the length of the right bracket is 1312mm, the inclination of the roller is 55 °; When the length of the right bracket is 1400mm, the inclination of the roller is 60 °; When the length of the right bracket is 1485 mm, the inclination of the roller is 65 °; When the length of the right bracket is 1570mm, the inclination of the roller is 70 °; When the length of the right bracket is 1652mm, the inclination of the roller is 75 °.

4: Page 7 line 196: According to the growing conditions of the sugarcane in the field, the sugarcane was clamped with a clamp to adjust to the corresponding lodging posture. Please describe the "corresponding lodging posture" in detail.

Authors: Thanks for the valuable comments. In test, to simulate the serious lodging of sugarcane, the lodging posture of sugarcane was set. The lodging angle of sugarcane was set as 10 °~20 °, and the the slip angle as 80 °~100 °.

Reviewer 2 Report

A variable spiral spike-tooth divider has been designed to address the problem of sugarcane lodging caused by tropical weather and climate. The design was tested by selecting the lifting angle of the sugarcane as the testing factor. The optimal range of test factors was determined using a single factor test, the relationship between the test index and the test factors was determined using an orthogonal test, and the regression model data was obtained by fitting the test data. Finally, the regression model was verified to be reliable by bench test. 

The research method of this paper is reasonable and the writing format is relatively standard, but there are the following problems

(1) There are semantic confusions and logical errors in the introduction of the article, How could the crop divider be a crucial component of the sugarcane? This statement may be changed to the fact that the divider is a crucial component of the sugarcane harvesting machine.

(2) In the introduction part of the article, there is little description of the own design of the divider, and there is no concise explanation of the advantages and improvements of the own design of the divider compared with the previous design. The focus could be adjusted accordingly.

(3) Some pictures and charts are better centered in the article.

(4) In the single factor test, the number of experiments can be increased in some. Thus, the pattern of sugarcane lifting angle variation is more credible and generalized.

Author Response

Dear Editor,

Based on the constructive comments, edits and suggestions provided by you and the reviewers, we have conducted a thorough revision of the manuscript. We have responded to all edits, comments and suggestions as discussed in the following pages in this document.

Please let me know if you have any questions or need further clarification.

Thank you very much for your time processing the manuscript.

Sincerely,

Shaochun Ma, Ph.D

Associate Professor

College of Engineering

China Agricultural University

No.17 Qinghua East Road, Haidian District, Beijing, 100083

Ph: 010-62736638; Cell: 18210198629

Email:[email protected]

Reviewer #2: A variable spiral spike-tooth divider has been designed to address the problem of sugarcane lodging caused by tropical weather and climate. The design was tested by selecting the lifting angle of the sugarcane as the testing factor. The optimal range of test factors was determined using a single factor test, the relationship between the test index and the test factors was determined using an orthogonal test, and the regression model data was obtained by fitting the test data. Finally, the regression model was verified to be reliable by bench test.

The research method of this paper is reasonable and the writing format is relatively standard, but there are the following problems:

1: There are semantic confusions and logical errors in the introduction of the article, How could the crop divider be a crucial component of the sugarcane? This statement may be changed to the fact that the divider is a crucial component of the sugarcane harvesting machine.

Authors: We are very grateful to Reviewer for reviewing the paper so carefully. It has been revised. (line 32, 33)

2: In the introduction part of the article, there is little description of the own design of the divider, and there is no concise explanation of the advantages and improvements of the own design of the divider compared with the previous design. The focus could be adjusted accordingly.

Authors: We are so grateful for your kind question. In picture, the crop divider with spiral catheter was developed by us. According to previous researches, the spiral blades of the crop divider were mostly formed by spiral catheter, and its continuous structure limited the contact mode with sugarcane. Therefore, the traditional crop divider was improved. The spiral catheter of the crop divider was replaced by the spike tooth, which were welded to the surface of the spi-ral scroll in a spiral manner. It can not only play the role of pushing function of the spiral lines, but also independently play the role of pulling function the spike tooth by referring to the pulling principle of the finger-chain type crop divider. The crop divider of spike-tooth can avoid the disadvantage of point contact of traditional spiral blades, which effectively solve the problem of sugarcane slipping and greatly improve the sugarcane lifting performance.

The crop divider with spiral catheter

3: Some pictures and charts are better centered in the article.

Authors: Thanks for the valuable comments. The location of pictures and charts has been changed.

4: In the single factor test, the number of experiments can be increased in some. Thus, the pattern of sugarcane lifting angle variation is more credible and generalized.

Authors: Thanks for the valuable comments. We agree that more experiments would be useful to the conclusion. The test was conducted in Guangxi Agricultural Machinery Research Institute. Because there is no test condition in Beijing, we feel very sorry that we cannot increase the number of experiments.

Reviewer 3 Report

In this study, to improve the lifting performance of crop dividers, a variable spiral spike-tooth crop divider was designed. Forward speed, the rotational speed of scrolls, and the installation angle of scrolls were selected as test factors, and the lifting angle of sugarcane was selected as the test index. The manuscript is not handled perfectly, especially the introduction and conclusion portions. The manuscript in its present form requires major revisons. Further comments and suggestions are given in attached file.

Author Response

Dear Editor,

Based on the constructive comments, edits and suggestions provided by you and the reviewers, we have conducted a thorough revision of the manuscript. We have responded to all edits, comments and suggestions as discussed in the following pages in this document.

Please let me know if you have any questions or need further clarification.

Thank you very much for your time processing the manuscript.

Sincerely,

Shaochun Ma, Ph.D

Associate Professor

College of Engineering

China Agricultural University

No.17 Qinghua East Road, Haidian District, Beijing, 100083

Ph: 010-62736638; Cell: 18210198629

Email:[email protected]

Reviewer #3: In this study, to improve the lifting performance of crop dividers, a variable spiral spike-tooth crop divider was designed. Forward speed, the rotational speed of scrolls, and the installation angle of scrolls were selected as test factors, and the lifting angle of sugarcane was selected as the test index. The manuscript is not handled perfectly, especially the introduction and conclusion portions. The manuscript in its present form requires major revisons. Further comments and suggestions are given in attached file.

1: Avoid start of sentence with And.

Authors: We are very grateful to Reviewer for reviewing the paper so carefully. It has been deleted. (line 19)

2: Background explained well but study gap and novelty is not explained properly.

3: directly move towards your work before study gap and this is very important paragraph try to improve it

4: Also Try to add the benefit and application of this study at the end of this paragraph

Authors: We are so grateful for your kind question. In picture, the crop divider with spiral catheter was developed by us. According to previous researches, the spiral blades of the crop divider were mostly formed by spiral catheter, whose continuous structure limited the way of contact with sugarcane. Therefore, the traditional crop divider was improved. The spiral catheter of the crop divider was replaced by the spike tooth, which were welded to the surface of the spiral scroll in a spiral manner. It not only exerts the pushing action of the spiral lines, but also the pulling action of the spike tooth independently with reference to the pulling principle of the finger-chain type crop divider. The crop divider of spike-tooth can avoid the disadvantage of point contact of traditional spiral blades, effectively solve the sugarcane slipping problem and greatly improve the sugarcane lifting performance. It was added. (line 54-62)

The crop divider with spiral catheter

5: As shown in figure 1 move at the end of sentence looks proper

Authors: Thanks for the valuable comments. It has been corrected. (line 66, 68)

6: Reference?

Authors: Thanks for the valuable comments. The reference of Formula 1:Cheng, Z.; Chen, W. Structure design & analysis and experiment research of the carrying frame in minitype sugarcane harvester. Master's Thesis, Guang Xi University, NanNing, 2008.” Formula 2: To prevent the sugarcane from slipping, it is necessary to ensure that the sugarcane can be lifted to a certain height, which should be greater than the height of the center of gravity of the sugarcane. Therefore, Equation 2 was obtained.

7: Define internodes ?

Authors: Thanks for the valuable comments. Sugarcane stems were selected to avoid the influence of other factors with relatively straight growth. Before the test, the sugarcane was stripped, and sugarcane was selected with roughly the same effective length and diameter.

8: Not just showed try to explaint the effect or losses in some sentences due to these three parameters.

Authors: We appreciate the reviewer for this kind recommendation. We did our best to analyze the reasons for the data trends in the graph.(242, 243; 251-254; 259-261)

9: font size of each graph sholud be same.

Authors: Thanks for the valuable comments. It has been corrected.(line 236)

10: Same like abstract mut be improved you have to mention the conclusions according to the results and give some key findings and factors. Also add at the end of the conclusion the application of this study and future perspective.

Authors: Thank you for your questions. Some key findings and future perspective were added. (line 336-339;344,345;352-356)

11: Add some recent literature also.

Authors: Thanks for the valuable comments.The recent literatures were added.

Round 2

Reviewer 3 Report

The author briefly explains my points and suggestions.  The manuscript in its present form is suitable for publication after some English corrections